# Electricity generation from digitally printed cyanobacteria

Marin Sawa [1,2], Andrea Fantuzzi [2], Paolo Bombelli[3], Christopher J. Howe[3], Klaus Hellgardt[4] & Peter J. Nixon [2]

Microbial biophotovoltaic cells exploit the ability of cyanobacteria and microalgae to convert light energy into electrical current using water as the source of electrons. Such bioelectrochemical systems have a clear advantage over more conventional microbial fuel cells which require the input of organic carbon for microbial growth. However, innovative approaches are needed to address scale-up issues associated with the fabrication of the inorganic (electrodes) and biological (microbe) parts of the biophotovoltaic device. Here we demonstrate the feasibility of using a simple commercial inkjet printer to fabricate a thin-film paper-based biophotovoltaic cell consisting of a layer of cyanobacterial cells on top of a carbon nanotube conducting surface. We show that these printed cyanobacteria are capable of generating a sustained electrical current both in the dark (as a 'solar bio-battery') and in response to light (as a 'bio-solar-panel') with potential applications in low-power devices.

[1] Central Saint Martins College of Arts and Design, University of Arts London, Granary Building, London N1C 4AA, UK. [2] Department of Life Sciences, Imperial College London, Sir Ernst Chain Building – Wolfson Laboratories, South Kensington Campus, London SW7 2AZ, UK. [3] Department of Biochemistry, University of Cambridge, Hopkins Building, Downing Site, Cambridge CB2 1QW, UK. [4] Department of Chemical Engineering, Imperial College London, Bone Building, South Kensington Campus, London SW7 2AZ, UK. Marin Sawa, Andrea Fantuzzi and Paolo Bombelli contributed equally to this work. Correspondence and requests for materials should be addressed to P.J.N. (email: p.nixon@imperial.ac.uk)

There is currently great interest in using living micro-organisms to produce an electrical current for use in 'green electronics'[1–3]. The main focus has for a long time been on the use of heterotrophic bacteria to convert organic carbon substrates into an electrical output in so-called microbial fuel cells (MFCs)[4, 5]. More recently, photoautotrophic cyanobacteria and unicellular algae have been successfully used to produce a minimal type of MFC, termed a biophotovoltaic (BPV) cell[6–8], that operates in the absence of an added carbon feedstock. Instead electrons are released in the light during the process of oxygenic photosynthesis and in the dark during the oxidation of carbohydrate or other carbon-containing compounds synthesised from carbon dioxide[8]. Thus, BPV devices are able to provide power in both the light and the dark, in contrast to photovoltaic (PV)[8] systems which are driven solely by light. Furthermore, BPV devices can repair light-induced damage to the photosynthetic apparatus, in contrast to semi-artificial PV systems containing isolated photosynthetic reaction centres[9–11], and are therefore more durable[12]. These features suggest that BPV devices could play a role as environmentally friendly power supplies for use in low-power applications.

Conventional BPV devices are made by gravity-induced deposition of cells from liquid culture onto an electrode surface[6, 8]. Such an approach has a number of drawbacks for scalability: the devices are relatively bulky due to the presence of a liquid reservoir, the sedimentation process is lengthy, and there is limited scope for precision engineering of the electrode[1, 13, 14].

Here we describe three innovations to improve the miniaturisation and large-scale production of BPV cells. Our approach is based on the use of inkjet printing which has been applied previously for the high-throughput patterning of various types of living cell onto solid supports[15–18] and is widely exploited for the deposition of sub-cellular components such as DNA and enzymes as well as the industrial-scale production of printed electrical conductors[19].

Firstly, we demonstrate the feasibility of using an inexpensive commercial inkjet printer to print a 'bio-ink' of cyanobacterial cells onto paper under conditions that allow the cells to remain fully viable and to retain their photosynthetic capacity after printing.

Secondly, we demonstrate that inkjet printing can be used to fabricate both the non-biological and biological parts of a 'bioelectrode' and that this printed bioelectrode produces an electric current at similar levels to the traditional bioelectrode used in BPV devices and is capable of powering a small digital clock or low-power LED light.

Finally, we use ink-jet printing to fabricate a 'thin-film semi-dry' BPV cell in which a water-absorbent gel is used to replace the cumbersome liquid reservoir. We show that this type of BPV device is capable of producing sustained current for more than 100 h.

## Results

**Digital printing of cyanobacteria.** The model cyanobacterium *Synechocystis* sp. PCC 6803 (hereafter *Synechocystis*)[20, 21] was chosen for our studies as it has been used extensively in BPV devices[22] and is amenable to metabolic engineering[12]. To help minimise clogging or damage to *Synechocystis* cells during printing, we decided to test a Hewlett-Packard (HP) Deskjet 340 inkjet printer which contains an ink cartridge with a 50 µm wide nozzle, one of the largest available commercially. In comparison, the average diameter of a typical coccoid *Synechocystis* cell is about 1.5 µm[23]. The thermal inkjet technology used by the HP Deskjet printer is also more benign for cell printing than piezo-electric inkjet technology[16, 24].

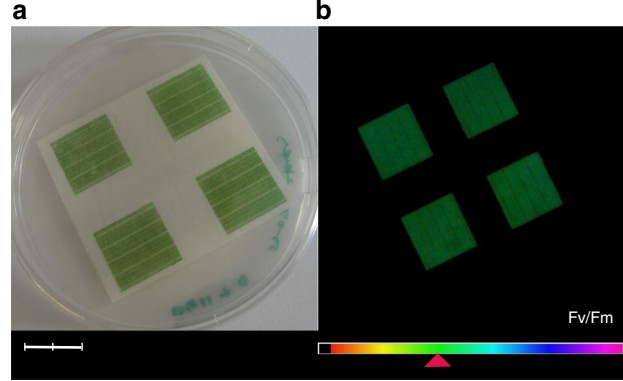

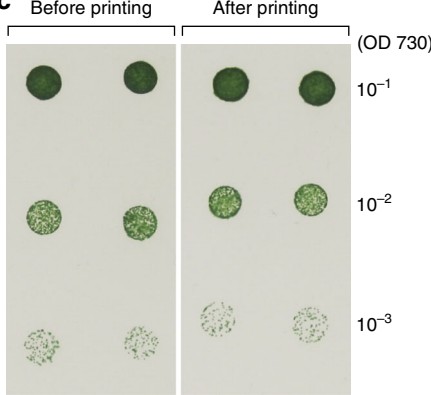

**Fig. 1** Cell viability and photosynthetic capabilities of digitally printed cyanobacteria. **a** Photograph of inkjet-printed *Synechocystis* cells after 3 days of incubation. Scale bar measures 2 cm. **b** Chlorophyll fluorescence image of the sample **a** by imaging PAM, showing maximum quantum efficiency of PSII (Fv/Fm) at the values of about 0.4 according to colour gradient in the legend bar. **c** The panel compares the growth of *Synechocystis* colonies before and after the inkjet printing process, following 5 days of incubation on a BG-11 agar plate. A 3 µl aliquot of cells from a dilution series representing $10^{-1}$, $10^{-2}$ and $10^{-3}$ of the original suspension was spotted. For the most dilute cell suspension taken after printing, $90.5 \pm 10.6$ colonies were counted, whereas $87.5 \pm 12.0$ colonies were counted before printing. The difference between these values was found to be not statistically significant (one-way ANOVA: $p = 0.815$) (Supplementary Table 1)

Initial experiments revealed that *Synechocystis* cells printed onto ordinary paper could be grown on top of an agar plate (Fig. 1a). Paper is increasingly considered as an attractive candidate for the development of disposable electronics due to the advantages of low cost, widespread availability, flexibility and environmental friendliness[2, 25]. Analysis of chlorophyll fluorescence using an imaging Pulse Amplitude Modulated fluorometer (PAM) confirmed that the incubated cells were photosynthetically competent. The maximum quantum efficiency of photosystem II (PSII), determined from the ratio of variable (Fv) to maximum (Fm) chlorophyll fluorescence (Fv/Fm), measured using single saturating light pulses, was found to be about 0.4 (Fig. 1b), in good agreement with values measured for cyanobacteria in liquid cultures (Fv/Fm = 0.3–0.5)[26, 27]. The printing process did not affect cell viability based on a comparison of the number of colony forming units before and after printing (Fig. 1c; Supplementary Table 1). The chlorophyll concentration of the printed cells on paper after the incubation was approximately 50 µg cm$^{-2}$, which is similar to that of a plant leaf[28].

Growth of printed *Synechocystis* cells on other porous (edible rice paper, nano-paper, woven fabric) and non-porous supports (inkjet coated plastic, indium tin oxide coated polyethylene

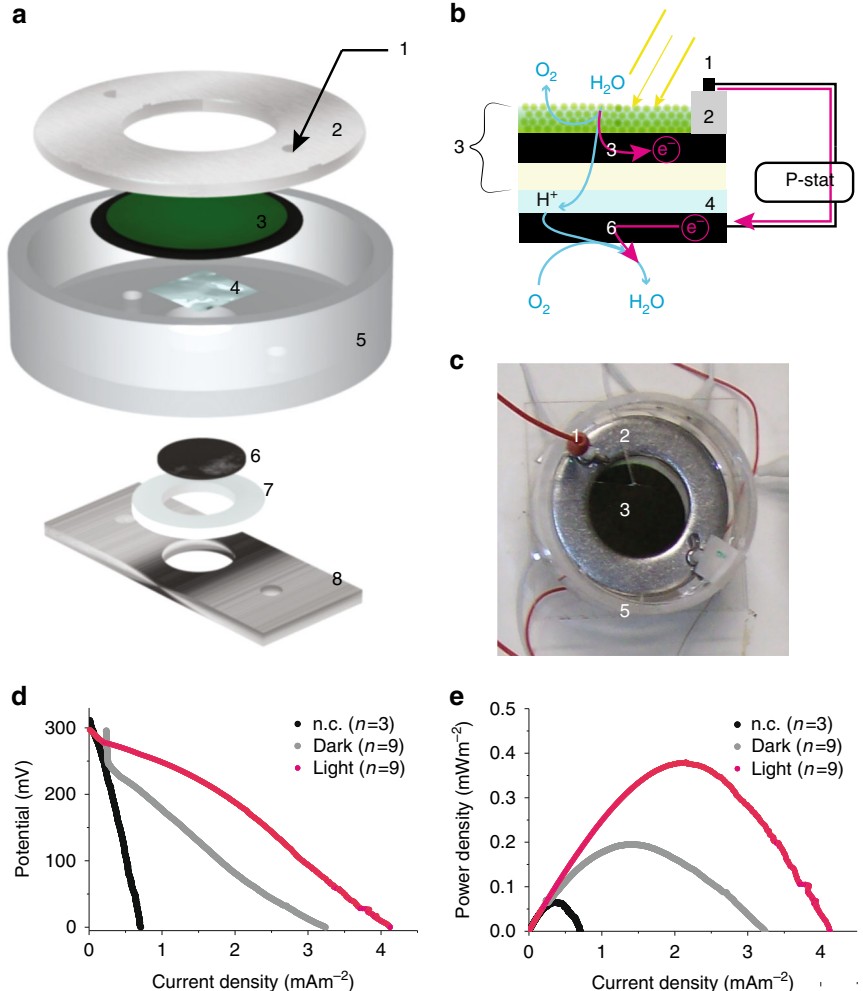

**Fig. 2** Electrochemical characterisation of a digitally printed bioanode in a hybrid BPV system. **a** Schematic representation (semi-exploded view) of the BPV unit with printed paper-based anode. Clamping screws (component 1); marine grade stainless steel ring for contacting the CNT anode (component 2); printed CNT anode in black (Ø 60 mm) with printed photosynthetic organisms in green (Ø 40 mm) with a total area of ~ 28.4 cm$^2$ (component 3); hydrogel (component 4); Plexiglas vessel (component 5); carbon paper-Pt, with a total area of ~ 3.5 cm$^2$ was used as cathode (component 6); silicon O-ring (component 7); stainless steel plate used to clamp all the component together (component 8). ~ 60 ml of BG-11 medium was placed above the printed cells in the chamber formed by the top plate. **b** Schematic representation of the BPV unit cross-section where electrons, protons and oxygen flow are also shown. Numbering as in **a**. **c** Photograph of the experimental setup (excluding the potentiostat and the wiring). Numbering as in **a**. **d** Polarization and **e** power curves for the printed anode in the BPV unit. Printed *Synechocystis* (incubated for 5 days after printing) on printed CNT anode exposed to light (magenta symbols) and in the dark (grey symbol) was compared with a bare printed anode (black trace). Number of repeats is indicated in parenthesis

terephthalate (ITO-PET)) was much poorer than on paper (Supplementary Figs. 1, 2), probably because these materials lack the microporous structure of paper required for high water absorption and the fibrous matrix needed for efficient wicking[25].

An alternative methodology based on pneumatic microvalve inkjet printers adopted in cell printing for biofabrication[18], was also tested (Supplementary Fig. 3). However, the larger volumes dispensed by this system led to over-wetting of the paper substrate, which hindered the precise and uniform deposition of the cyanobacterial cells on the paper support.

**Construction and characterisation of a digitally printed bioelectrode**. To test the electrogenic properties of printed cyanobacteria, we fabricated a bioelectrode, which is defined as the combination of photosynthetic organisms with an inert electrode material[29]. The cyanobacterial bioelectrode was printed in a two-step process: firstly, the electrode was printed on the paper substrate using an inorganic conductive inkjet ink and secondly the cyanobacteria were printed onto the electrode pattern on the paper. The conductive inkjet ink we chose was the "Nink-1000: multiwall" (NanoLab, USA), which consists of carbon nanotubes (CNTs) in aqueous suspension. CNTs have previously been used as conductive patterns on paper substrates[30, 31] and have been shown to be compatible with the growth and electrochemical analysis of cyanobacteria[32]. We found that 5 to 6 overlays of the Nink-1000 conductive ink could be printed to give a conductive surface with a resistivity in the range of 5–10 kΩ cm and that cyanobacteria could be printed and grown directly on the CNT electrode on paper (data not shown).

In order to compare the performance of our printed bioelectrode to the bioelectrode formed by gravity-deposition of cells[22], characterisation was carried out by forming a 'hybrid' biophotovoltaic cell consisting of the printed bioelectrode paired with the platinised carbon cathode electrode used in conventional BPV devices. Figure 2a–c illustrates the assembled hybrid BPV cell in which the cathode is exposed to the air as in the conventional BPV system.

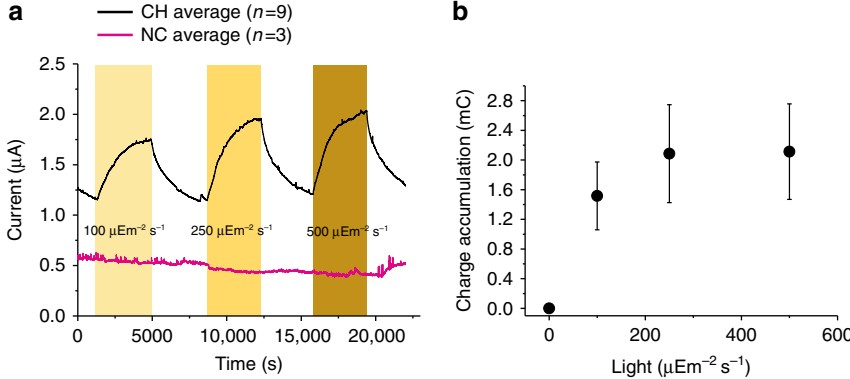

**Fig. 3** Effect of light intensity on anodic photocurrent produced by the hybrid BPV system. **a** Current output measured over 6 h with three periods of light and darkness (1 h each). Light periods indicated by the yellow bars. Black trace for inkjet-printed *Synechocystis* on printed CNT anode and magenta trace for control experiments without the printed cells. Number of repeats is indicated in parenthesis. **b** Saturation curve for the current outputs as presented in the **a**. For each period of light (100, 250 and 500 µE m$^{-2}$ s$^{-1}$), the current was integrated over time. The charge attributable to dark current over the same time was subtracted from the total charge during the light periods and plotted vs. the photon flux. Each data point is the result of 9 replicates and the standard error is shown as error bars

Polarisation curves (Fig. 2d) were used to characterize the printed *Synechocystis* bioelectrode and were recorded by performing linear sweep voltammetries (LS) in the absence and in the presence of light (100 µE m$^{-2}$ s$^{-1}$). The maximum current density output generated by the printed cells was found to be just over 4 mA m$^{-2}$ in the light and 3 mA m$^{-2}$ in the dark (Fig. 2d). This range is approximately 3 to 4-fold higher than previously recorded (ca. 1 mA m$^{-2}$) for the same *Synechocystis* strain deposited on an ITO-PET electrode using the conventional approach[22].

Power curves (Fig. 2e), derived from the polarisation curves using Ohm's law, showed a clear effect of light with a peak power output of $0.38 \pm 0.07$ mWm$^{-2}$ and $0.22 \pm 0.07$ mWm$^{-2}$ in the light and in the dark, respectively, a difference that was found to be statistically significant (one-way ANOVA: $p < 0.0005$) (Supplementary Table 2). This range is again 3 to 4-fold higher than previously recorded (ca. 0.12 mW m$^{-2}$ with a slight difference between the dark and light cycles)[22].

In the absence of cells, the peak power output was considerably lower $(0.07 \pm 0.01$ mW m$^{-2}$; Fig. 2e, black symbols) and insensitive to the presence of light (data not shown). The difference in power output in the dark with and without the cells on the anode was found to be statistically significant (one-way ANOVA: $p = 0.005$) (Supplementary Table 3).

The printed system was characterised further by chronoamperometry, which monitors the current output as a function of time and records changes induced by external stimuli such as light. The chronoamperometric experiments were performed at three different light intensities (100, 250 and 500 µE m$^{-2}$ s$^{-1}$) separated by periods of 1 h in the dark. Increases in current output were observed only in the samples with the printed cyanobacteria while no changes were observed in the controls (Fig. 3a). As shown in Fig. 3a the currents measured in the presence of light were higher than in the dark and their magnitudes were comparable to the ones measured in the potential scanning experiments (Fig. 2d). Figure 3b shows the values of the total charge accumulated as a function of the intensity of the light, calculated by integrating the current output over time and subtracting the contribution of the dark current. The device exhibits the expected light saturation of the photosynthetic apparatus[12] with saturation of the current output observed at light intensities above 200 µE m$^{-2}$ s$^{-1}$.

**Powering a digital clock**. To assess the ability of the printed bioelectrode to power a small electronic device, such as a biosensor, we tested whether the hybrid BPV unit shown in Fig. 2 could power a digital clock. This test allows direct comparison with literature reports where conventional BPV devices have been shown to power a digital clock when connected in series[22].

Nine replicates of the hybrid BPV unit were arranged in three clusters connected in parallel. Each cluster had three units connected in series (Fig. 4a). This setup produced an overall voltage output of 1.4–1.5 V and an overall current output of 1.5–2 µA, a good compromise based on the clock manufacturer's specifications for powering the digital clock.

We found that the digital clock was successfully powered by the BPV array for 'ON' periods of 30 min alternated with 'OFF' intervals of 30 min to allow the BPV devices to recover (Fig. 4b). The chronovoltammetry and chronoamperometry curves in Fig. 4b clearly indicate the discharge of the array when connected to the clock to activate it ('ON'): there was a rapid decrease followed by stabilisation of the voltage. On disconnection, there was a rapid increase in voltage followed by stabilisation of the potential across the anode and cathode. This process was repeated several times demonstrating reproducibility.

**Powering a LED**. Small and low power electronic devices such as biosensors often work over short measuring periods, interspaced by longer periods of inactivity. To assess the ability to generate a relatively high power output in short bursts, we tested whether Hybrid BPV units could generate flashes of light from an LED (Fig. 4c).

The LED was connected to a pulse generator whose electronic scheme is presented in the supporting information (Supplementary Fig. 4). To generate the required voltage (ca. 3 V), an array consisting of 9 Hybrid BPV cells was connected in series (Fig. 4c), so that the output voltage is the sum of the 9 units. To accumulate the required charge, the BPV array was charged for 1 h, then the circuit was closed for 60 s during which the LED was pulsed at a frequency of one pulse every 2.5 s. We detected in ten separate experiments an average of 24 flashes in this 60 s period, confirming that the BPV array could indeed generate bursts of power sufficient to drive the LED (Fig. 4d).

Immediately after closing the circuit a short spike of current intake (~ 35 µA) and potential drop (~ 1.5 V) lasting ~ 2 s were observed (Fig. 4d). This vigorous initial electrical consumption is an expected phenomenon due the circuit capacitance and a similar behaviour was also observed with the digital clock

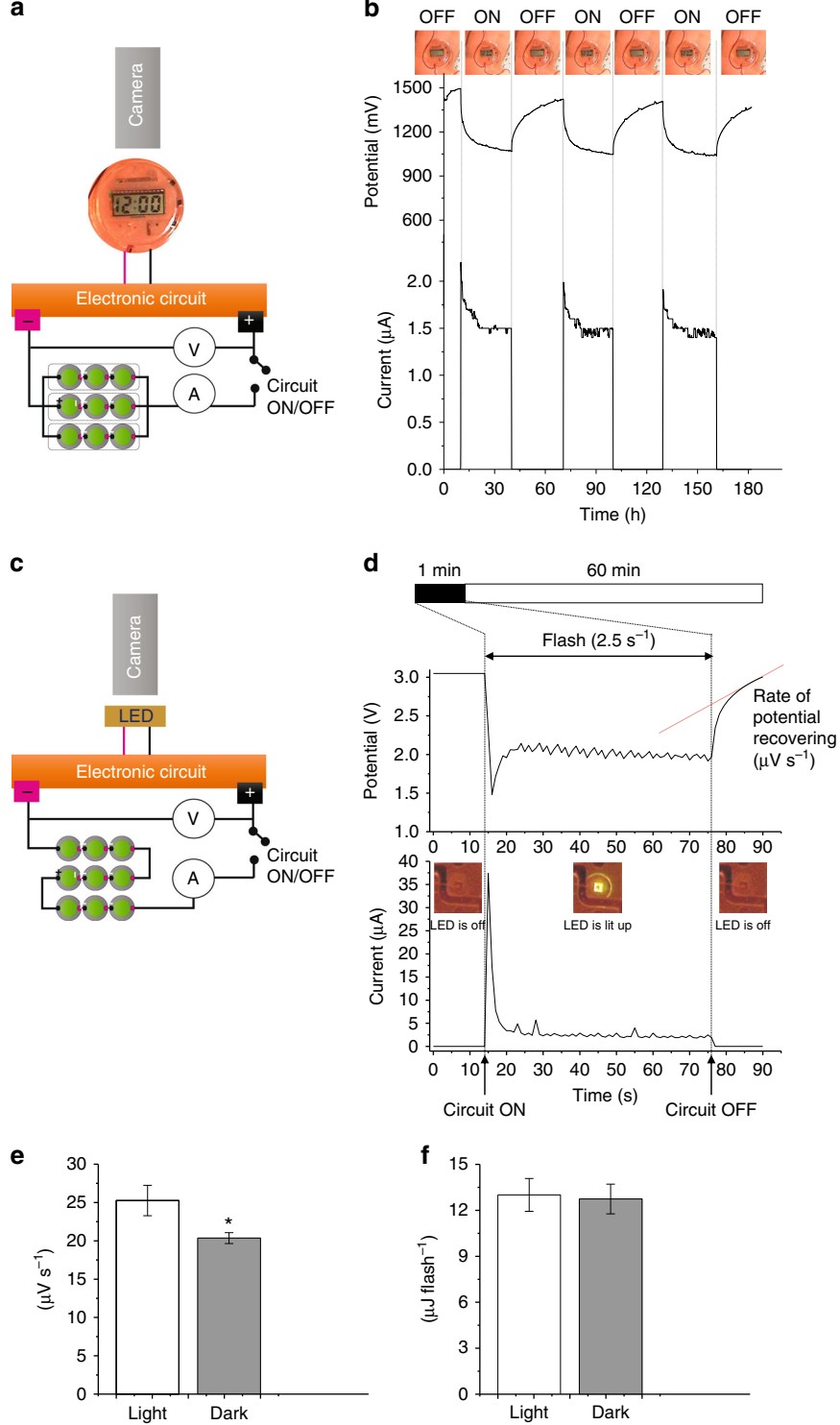

**Fig. 4** Powering a clock and a LED-flash with an array of Hybrid BPV units. **a** Schematic representation of the experimental setup for the powering of a digital clock. An array consisting of 9 Hybrid BPV cells were organised in 3 clusters connected in parallel. Each cluster had 3 units connected in series. **b** Chronovoltammetric and chronoamperometric traces recorded during the experiment where the circuit (*i.e.*, the digital clock) was either on (*i.e.*, clock activated) or off (*i.e.*, clock deactivated) for periods of approximately 30 min. **c** Schematic representation of the experimental setup for the powering of a LED. The array was organised all in series. **d** Chronovoltammetric and chronoamperometric traces recorded during the experiment where the circuit with its integrated LED was either on (*i.e.*, pulsing every 2.5 s to activate the LED) for periods of approximately 60 s or off (*i.e.*, LED deactivated) for periods of approximately 1 h. Rate of voltage recovery was estimated by fitting the last 7 s of data with a linear regression line (in red); **e** kinetics of recovering to the original voltage following LED pulse when the BPV array was kept in the dark and when it was exposed to light; **f** average energy consumed for each LED pulse

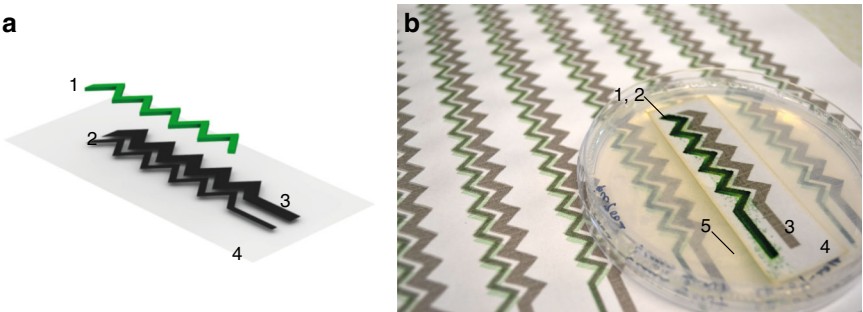

**Fig. 5** Design of a fully printed BPV system. **a** Schematic representation (semi-exploded view) of the digitally printed bioelectrode module. 1: Printed photosynthetic organisms in green; 2: Printed CNT anode; 3: Printed CNT cathode; 4: Paper substrate. The one module consists of one zigzag anode and one zigzag cathode with surface areas 1.36 cm$^2$ and 2.73 cm$^2$, respectively. **b** Photograph of A4-size arrays with freshly printed *Synechococcus* cells, compared to the incubated module grown on an agar plate for 3 days. (Note the enhanced green colour of the growing cyanobacteria.) 1–4 are the same as **a** and 5 is the solid medium

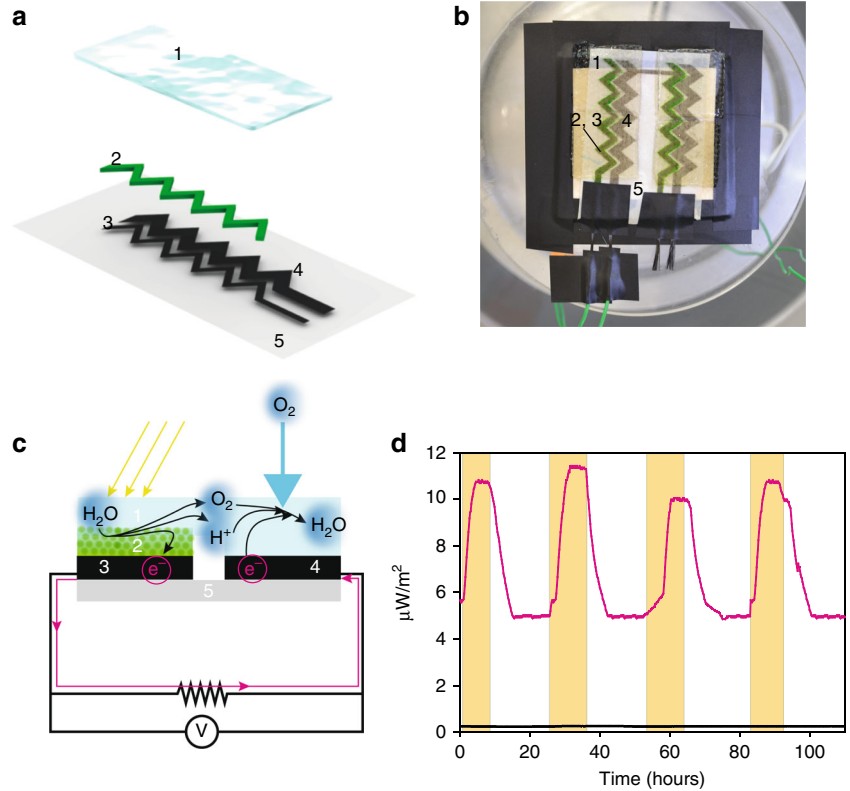

**Fig. 6** Testing the performance of the fully printed BPV device. **a** Schematic representation (semi-exploded view) of the printed paper-based BPV cell. Paper support in light grey (component 5); Printed CNT anode (component 3) and CNT cathode (component 4) in black; Printed *Synechocystis* in green (component 2); Bridging hydrogel in pale blue (component 1). **b** Photograph of the experimental setup (excluding the potentiostat), showing a pair of BPV modules printed in series. **c** Schematic representation of the BPV cross-section where electron, proton and oxygen flows are also shown. **d** Power output measured over 4 days with periods of light and darkness. Light periods indicated by the yellow bars. Magenta trace for inkjet-printed *Synechocystis* on printed CNT anode and black trace for control experiments without the cells

(Fig. 4b). For the remaining 58 s, the current driven by the board stabilised at around 2–3 μA with a closed-circuit potential of ~ 2 V.

Recovery of the voltage following the LED discharge was also monitored and its kinetics were measured when the BPV array was kept in the dark and when it was exposed to light. Figure 4e shows that in the presence of light (100 μE m$^{-2}$ s$^{-1}$) the recovery was faster suggesting that the photosynthetic reactions in the cyanobacteria accelerate the recovery of the system following discharge. Fig. 4f shows the energy consumption at each flash of

the LED light. The dark/light regime did not cause any significant variation in the energy consumption. As expected, the energy delivered by the pulse generator was independent of the activity of the BPV array, as the energy (~ 13 μJ) was delivered to the LED only when the capacitor of the pulse generator was charged.

**Design and testing of a semi-dry thin-film BPV system.** Having demonstrated the ability of the printed bioelectrode (as a bioanode) to generate a sustained power output in a hybrid BPV

system, we fabricated a printed BPV cell where not only the anode but also the cathode was patterned and printed on paper. Power output in microbial fuel cells strongly depends on the overall dimensions of the electrodes and the relative distance between the planes of the anode and the cathode[5, 33]. In order to maintain a compact size, we designed a zigzag electrode pattern with a view to enabling the conduction of ions between anode and cathode while reducing the overall length of the electrodes. Furthermore, the cathode surface area was made larger than that of the anode in order to enhance the exposure to oxygen and reduce catalytic limitations[5, 33].

As before, a two-step inkjet printing process was used to make the printed BPV system with the electrogenic cyanobacteria deposited precisely on the zigzag anode (Fig. 5). Our tests confirmed that 5 to 6 overlays of the ink printed gave a conductive surface with a resistivity of the order of 100 kΩ cm. This value is higher than that observed previously (Fig. 2) due to variability between different batches of the Nink-1000 ink. Nevertheless, *Synechocystis* (data not shown) as well as the related cyanobacterium *Synechococcus* sp. PCC 7002 could be printed and grown directly on the conductive ink (Fig. 5).

The printed bioelectrode module made with *Synechocystis* cells was assembled into a thin film configuration by covering the anode and cathode with a water-absorbent gel (hydrogel) so as to form a slim BPV construct. Hydrogels have previously been used to encapsulate microorganisms in MFCs[34, 35], but in our application the hydrogel covering the biofilm functions both as a salt bridge connecting the anode and cathode and as a supply of minimal growth medium and water to the printed cells (Fig. 6). It plays an equivalent role to the bulky liquid reservoir in conventional BPV systems and, together with the paper-based solid culture system, reduces the volume of the BPV device substantially. As indicated in the Methods section, the assembled system was then placed into a chamber where humidity and illumination were controlled to prevent dehydration.

As this configuration was difficult to connect to an external reference electrode, the electrochemical characterisation was carried out with a voltmeter and an external load[33]. The recorded power output (Fig. 6d) was found to be smaller than that measured in either the hybrid BPV system of Fig. 2 or the literature values of a conventional BPV system[21]. The reduction could be due to an increase in the internal resistance of the printed circuit since the magnitude of the decrease in the power output matched the increase in the ink resistivity

Long-term stability of the power output was assessed over a 10 h light/14 h dark cycle for 4 days (Fig. 6d). During each light/dark cycle, the peak power reached a maximum within two to three hours and then remained stable until the light source was switched off, after which the power returned to the stable value recorded in the dark. Overall, power output was observed to be stable (within a 10 % error) during the light/dark cycles throughout the period of over 100 h.

## Discussion

We have described here a radically different approach for the construction of BPV cells that uses inkjet printing to print digitally both the cyanobacterial and the electrode components. The cyanobacterial cells survived the printing process and were able to grow on the printed electrode to form a solid culture (Figs. 5, 6) which helps reduce the volume of starting culture thereby improving water-use efficiency, an important consideration when scaling up the growth of cyanobacteria[36]. Growing a solid layer of cyanobacteria allowed us to replace the liquid reservoir normally used in conventional BPV devices with a gel to

form a 'semi-dry thin-film' BPV cell, which has potential for miniaturisation as well as opening new avenues for large-scale production.

Detailed testing of the printed bioelectrode in a hybrid BPV cell revealed a power output of 0.38 mW m$^{-2}$ in the light and 0.22 mW m$^{-2}$ in the dark (Fig. 2). This power range compares well with previous results obtained with conventional liquid culture-based BPV devices (0.2–0.3 mW m$^{-2}$)[22].

Power in BPV devices is generated by photosynthetic reactions in the microorganisms printed on the anode and, unlike traditional MFCs, does not depend on the concentration of organic metabolites in the anodic compartment. Although power output from the semi-dry thin-film BPV device was less in the dark than in the light, it was stably maintained for several hours once the light was turned off. This is a general phenomenon observed with photosynthetic organisms in BPV devices (as seen in Figs. 2, 3 and 6) and is probably due to the fact that the current can still be generated from the metabolism of internal storage reserves produced via photosynthesis during the illumination period[12]. Therefore, such a device would work as a bio-solar panel during the day and as a bio-battery during the night.

We showed that electrical output from the printed BPV cell can be sustained in light/dark cycles for > 100 h (Fig. 6), a distinct advantage over paper-based MFCs which can only operate for ~ 1 h probably because of diffusion of the anolyte and catholyte through the paper[37].

Given the current maximum power output from our digitally printed bioelectrode (0.38 mW m$^{-2}$), our printed BPV device holds most promise for low-power applications such as biosensors that use between 10 and 100 µW at 1–2 V, though efforts are currently directed at decreasing the power consumption of disposable biosensors[38, 39]. Realistic areas of use would be to provide power for point-of-care diagnostic devices where only a short burst of power is needed in the detection phase[37]. We have confirmed the feasibility of such an application by demonstrating the capability of our system to deliver sustained power to a digital clock and to deliver a burst of relatively high power to flash a LED by connecting 9 modules of the hybrid BPV cell in series and/or in parallel (Fig. 4). Furthermore, the long-term stability reported here would be suitable for either a small power supply or the sensing layer for environmental monitoring based biosensors[40–42].

The paper-based thin-film BPV cell described in Fig. 6 might form the basis of a disposable and environmentally friendly power supply for use in paper-based analytical devices (PADs), which have attracted considerable attention for point-of-care applications by combining the advantages of low cost and ease of use with sensitivity, specificity, robustness and disposability[43–45]. We can therefore envision future applications where PADs, disposable electronics and paper-based thin-film BPV power supplies are fully integrated into a single biodegradable paper-based lab-on-a-chip.

From a design perspective, the potential scalability and creativity of this digitally printable paper-based bioelectricity device suggest that much larger print sizes or module systems could be developed including, possibly, bioenergy wall paper (Supplementary Fig. 5).

There is still considerable potential for enhancing the power output of our system. For the non-biological parts this could include: improving the catalytic performance of the printed CNT cathode, which is considered a limiting factor in microbial fuel cell performance[33] and which is currently an area of intensive research;[46] increasing the circuit's conductivity by using more conductive metal or non-metal based inkjet inks[19], combined with insulating the paper-based circuit tracks with inkjet-printable hydrophobic polymer[47], and by optimising cell design. For instance a sandwich structure (Supplementary Fig. 6), with

the hydrogel located between a parallel anode and cathode, would improve the power output by exposing the cathode to more air[33], and increasing the electric field and ion mobility[33]. Furthermore, in order to limit water evaporation from the hydrogel, the addition of a gas-permeable membrane represents a simple solution.

There is also scope for improving the biological part. The use of cyanobacteria and algae of greater electrogenic potential[22] and/or greater resistance to dehydration[48] and the damaging effects of high light (such as desert cyanobacteria[49]) might both improve electron production and drastically decrease the need for hydrogel, agar, and humidity control, thereby reducing further the material and energy costs of scale-up.

## Methods

**Cell preparation and bioink cell suspension.** *Synechocystis* sp. PCC 6803 was used for the electrochemistry experiments and for the viability test and the glucose-tolerant wild type (WT-G)[50] for the imaging-PAM chlorophyll fluorescence experiment. The WT-G strain was grown in BG-11 medium[50] and *Synechocystis* PCC 6803 in BG-11 medium containing 3.6 % (w/v) NaCl (BG-11 high salt) until mid-log phase ($OD_{730}$ of 0.25 measured using a Shimadzu UV-1601 spectrophotometer (Shimadzu, Japan)), pelleted by centrifugation and resuspended in $1/100^{th}$ the volume of fresh BG-11 medium. The concentrated cell resuspensions were reconstituted to form a 'bioink' in a Falcon tube and kept in the container till before the printing process. For the growth experiment in Fig. 5, a liquid culture of the cyanobacterium *Synechococcus* sp. PCC 7002 was grown in medium A[49] supplemented with D7 micronutrients[51], and cells concentrated as above. Agar plates contained BG-11 medium supplemented with 1.5% (w/w) agar. Cells were grown at 30 °C at an irradiance of 20–30 $\mu$E m$^{-2}$ s$^{-1}$ of fluorescent white light (Sylvania Gro-Lux tubes). Copy paper, white, A4, 80 g m$^{-2}$ purchased from OfficeDEPOT was used without any coating or manipulation.

**Printing cells.** Hewlett-Packard (HP) Deskjet 340 and HP 33 ink cartridges were used for the printing of the cyanobacteria without any modification. The cartridge was emptied of the ink, cleaned and sterilised by rinsing with deionised water and ethanol as described by Wilson & Boland[15]. The pre-emptied sterile cartridge was then filled with the bioink using a micropipette. The filled cartridge was left to stand for up to 10 min to stabilise the inside air pressure. Following this process, it was inserted into the printer device. The inside of the printer was sterilised with ethanol where possible. Sheets of the copy paper were microwaved in protective plastic sleeves for 3–5 s for sterilisation and were fed in the printer. Prepared patterns in PDF were printed from a computer connected to the printer. The software drivers of the printer were used without any modification.

The first few prints were made to remove any remaining water (from the cleaning) in the pressure chamber and printhead. The printed cells on paper were then left to dry in the air in ambient conditions and the 'print' was transferred to the agar plate within an hour of printing. For growth, the plate was then incubated.

**Printing a bioelectrode.** The printed electrode circuit pattern was designed using graphic design software *Adobe Illustrator* (Adobe Systems, USA). Firstly, CNTs were printed in five over-layers to prepare a bespoke electrode circuit on paper, with anode and cathode electrodes with a size ratio of 1:2. Secondly *Synechocystis* cells were printed in five overlays onto the printed anodic areas (Fig. 6). The same inkjet printer, HP Deskjet 340, was used to print the electrode circuit, using the multiwall carbon nanotubes (MWCNTs) inkjet ink, Nink-1000: multiwall (NanoLab, USA), and the photosynthetic cells, *Synechocystis* PCC 6803 or WT-G. BPV printed modules consisted of one zigzag anode and one zigzag cathode with surface areas 1.36 cm$^2$ and 2.73 cm$^2$ respectively (Fig. 6a). For the stability experiments two printed BPV modules with the zigzag pattern were connected in series. The freshly printed cells on the BPV modules were transferred onto a BG-11 agar plate within an hour of printing and were incubated at 30 °C under continuous illumination of 10 W m$^{-2}$ s$^{-1}$ from white fluorescent lamps for 3 to 4 days.

**Cell viability analysis.** Cell viability of printed *Synechocystis* PCC 6803 was analysed by comparing the numbers of colonies before and after printing. The culture was grown in BG-11 high salt medium, and cells pelleted and resuspended to form bioink (a concentrated solution of cells in the medium) using the aforementioned method. Using the same printer and ink cartridge, as described earlier, the bioink was digitally printed onto a microscopic glass cover slip 22 × 22 mm Deckgläser (Menzel Gläser, Germany) and printed cells were immediately resuspended with pipetted distilled water and collected as a solution. The $OD_{730}$ of the suspension was measured to be 0.066 using a Shimadzu UV-1601 spectrophotometer. The suspension of cells before printing was sampled from the unused bioink and its $OD_{730}$ was adjusted to that of the suspension of printed cells. The cell suspension

was serially diluted (x 0.1, × 0.01, × 0.001) (Fig. 1c) and a 3 $\mu$l droplet from each of the suspensions was spotted onto solid medium (BG-11, agar 1.5 % w/w) and incubated at 30 °C under continuous illumination of 40 $\mu$E m$^{-2}$ s$^{-1}$ from white fluorescent lamps. Chlorophyll was extracted and amount determined as described by Porra et al[52].

**Hybrid and printed BPV systems for testing the printed bioelectrode.** The inkjet-printed paper-based bioelectrode was tested in two different systems: one with printed bioanode paired with platinised carbon cathode (hybrid BPV system) and the other with both printed anode and cathode (printed BPV system).

As shown in Fig. 2, the printed bioelectrode (cells and a CNT-based electrode) was prepared in the aforementioned way, and was used as a 'bioanode' placed above a cathode made of carbon paper-Pt (Johnson Matthey Company, USA). The printed bioanode had a total geometrical area of 12.5 cm$^2$. The cathode had a total geometrical area of 3.5 cm$^2$. The anode and cathode were assembled in such a way to avoid direct contact to prevent short-circuiting. Marine-grade stainless steel metal mesh (Mesh Company Ltd, UK) was used as contacts for the cathode. Marine-grade stainless steel ring (Ø 40 mm and 85 mm internal and external diameter respectively) was used for contacting the printed bioanode. Plexiglas vessel (Ø 90 mm and 100 mm internal and external diameter respectively) was used to create the anodic chamber. ~ 60 ml of BG-11 medium was placed above the cells in the anodic chamber formed by the plexiglas vessel. A silicon O-ring and a stainless-steel plate used to clamp the cathodic components together with the anodic chamber. Two plastic dielectric M5 screws were used to keep all the components of the BPV system firmly in place.

The printed BPV system was assembled by first incubating a printed bioelectrode, consisting of both anode and cathode, on the agar plate for 4 days and then by removing it from the plate and covering it with a hydrogel film over both anode and cathode. The hydrogel was about 1 mm thick containing the medium (BG-11 high salt), and acted as a salt bridge between the anode and cathode. It was pre-soaked in the appropriate aqueous medium. A commercially available hydrogel (Spenco®2nd Skin Squares, Spenco Medical Corporation, USA) was used and was made of superabsorbent polymer holding at least 80 % water. The hydrogel was transparent, allowing unimpeded illumination over the surface of the deposited photosynthetic cells. The electrodes' contacts were dried, cleaned and connected with carbon connectors. The contacts were insulated to avoid interference from humidity during the measurements. The assembled system was then placed inside a spherical aquarium *biOrb 60* (Reef One, UK) where 100 % relative humidity and white LED illumination (50 $\mu$E m$^{-2}$ s$^{-1}$) were controlled to prevent dehydration.

**Powering a digital clock and LED.** The digital clock and its attached electrical circuit were obtained from 4M Industrial Development Ltd. To power the digital clock, an array consisting of 9 Hybrid BPV cells (Fig. 2a) was organised in 3 clusters connected in parallel. Each cluster had 3 units connected in series.

The ultra-low current LED flasher was fabricated by the Electronics Workshop, Department of Psychology, University of Cambridge by following the blueprint available in the supplementary material (Supplementary Fig. 4). To power the LED flasher, the array of the 9 BPV cells were organised in series. A pulse generation circuit was placed between the LED and the BPV array.

In both cases, a voltmeter and an ammeter were used to measure the potential (Volt) and the current (Ampere) respectively (UNI-T, UT70B). Constant illumination was provided throughout (100 $\mu$E m$^{-2}$ s$^{-1}$) (OSRAM, LED 2700 K Warm White).

**Electrochemistry.** The two systems were tested for current production:

For the hybrid BPV system, polarisation curves (Fig. 2e) were recorded by performing linear sweep voltammetries (LS) in the absence and in the presence of light (100 $\mu$E m$^{-2}$ s$^{-1}$), from the voltage equivalent to the open circuit potential, to zero mV, as typically used for the characterisation of MFC[53]. Measurements were carried out with an Autolab PGSTAT12 (Metrhom/EcoChimie, the Netherlands) connected to a computer. Current (Fig. 3) was measured by connecting the BPV device to a potentiostat (P-stat, MultiTrace PalmSens) with a two-electrode system and setting the voltage to 0 V in order to measure the maximum current output.

For the printed BPV system, current (Fig. 6d) was measured with a UT60-A Digital multimeter (Uni Trend Group Ltd, China) with a 1 M$\Omega$ load connected in parallel (see Fig. 6b). Illumination was provided by white LEDs (50 $\mu$E m$^{-2}$ s$^{-1}$) and was placed approximately 50 cm above the surface of the printed BPV at which point light intensity was measured.

**Statistical validation.** The statistical analysis tool one-way analysis of variance (ANOVA) was used to determine whether there were any significant differences between the means of independent (unrelated) groups of data. When the *p*-value is greater than 0.05 there is no statistically significant difference between group means. The complete results obtained from the ANOVA tests run in this study are shown in Supplementary Tables 1-3.

**Data availability**. All relevant data are available from the authors upon reasonable request.

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

## Acknowledgements

We are grateful to Prof. Alexander Ruban and Dr Petra Ungerer at Queen Mary University of London (QMUL) for help with the Imaging-PAM experiment. We particularly thank our colleagues Dr Jianfeng Yu and Dr Hussein Haji Taha for providing some of the cultures and Dr Jianfeng Yu and Shengxi Shao for help with the cell viability experiments and analysis. M.S. is grateful to the University Arts London Central Saint Martins College of Arts and Design for the award of an International Graduate Scholarship and to James Swinson for his support and encouragement. The authors are grateful for funding provided by the UK Engineering and Physical Sciences Research Council (EPSRC), EnAlgae (http://www.enalgae.eu/, INTERREG IVB NWE) the Shuttleworth Foundation, and the Leverhulme Trust.

## Author contributions

M.S., with contributions from P.J.N. and K.H., conceptualised the idea of using an inkjet printer to print cyanobacteria and performed the printing experiments. M.S. and A.F. designed the thin-film BPV assembly. M.S., A.F. and P.B. devised and performed the electrochemical experiments. M.S., A.F., P.B., C.J.H., K.H. and P.J.N. analysed the data and drafted the manuscript. All authors read and approved the manuscript.

## Additional information

**Competing interests:** The authors declare no competing financial interests.

