## [Peer Review File · Nature Communications]

Reviewers' comments:

Reviewer #1 (Remarks to the Author):

1. Summary of the key results

The study presents a new design of paper-based BPV cell, which is able to digitally print live cyanobacterial cells on the electrode surface and generates electrical current. The formation of a solid culture is different from traditional liquid culture.

2. Originality and interest: if not novel, please give references

While the concept is interesting, and novelty of the study can be better justified. The inkjet method for bacteria deposition has been reported before, and it is not clear what's new in the deposition method for cyanobacteria in this study. An example study is "Xu, et al. Construction of high density bacterial colony arrays and patterns by the ink-jet method. *Biotechnology and Bioengineering*, 2004, DOI: 10.1002/bit.10768).

Similarly, using bioelectrochemical systems for biophotovoltaic applications have been reported and recently reviewed, as the author mentioned. (see McCormick et al., *Biophotovoltaics: oxygenic photosynthetic organisms in the world of bioelectrochemical systems*, *Energy & Environmental Science*, 2015, DOI: 10.1039/C4EE03875D). It is not clear what advantages this study has and why using solid culture is novel, applicable, and sustainable. More specifics are needed.

3. Data & methodology: validity of approach, quality of data, quality of presentation

The manuscript mainly discusses the manufacturing process of the biophotovoltaic cells and the process of experiment, but it has very limited information on scientific data reporting, characterization, or discussion. Plus, it is not clear what the research hypothesis is and what scientific questions are answered by the study.

The experimental design and reporting can be improved significantly. For example, it would be very informative if the authors connect different parameters such as light density, biomass growth, and power production and quantify their relationships and discuss how to improve system performance.

4. Appropriate use of statistics and treatment of uncertainties

ANOVA test was used to characterize the viability of data.

5. Conclusions: robustness, validity, reliability

The conclusions can be better supported by more comprehensive data and characterization. Please see comments above.

6. Suggested improvements: experiments, data for possible revision

Below are some specific comments and suggestions to help improve the quality of the study.

Page 8 - The compact gel to replace liquid reservoir growth media can be interesting, but can you provide data on what rate the gel substrate is consumed, how to replenishing the gel substrate, and why this is better than liquid media?

Page 10 - Figure 3. The power outputs are among the lowest reported by miniature microbial fuel cells, presumably due to the extremely large resistance present in the system and limited bacterial growth. Can you elaborate more on how did the configuration and light exposure affect power outputs, what are the limiting factors, and how to improve and sustain system performance?

Page 11 - Figure 3. The anode only control shows pretty significant power output relative to the anodes with cells. Can you explain why?

Page 18 - Can you provide more details on how the bioink was made and maintained?

Page 19 - How was the anode and cathode arranged to prevent short circuiting? Any separators applied?

Page 21 - the 1M Ω resistor used in the study seems too big, because it significantly limits electron

transfer between the anode and cathode. Most MFC studies use 10-1000 Ω to allow bacteria to transfer electrons.

7. References: appropriate credit to previous work?

The authors provided appropriate credits to previous work.

8. Clarity and context: lucidity of abstract/summary, appropriateness of abstract, introduction and conclusions

The writing is understandable, but really the science is missing.

Reviewer #2 (Remarks to the Author):

This paper deals with biophotovoltaic systems and more precisely with the demonstration of a device that generates electrical power from microbacteria. The main novelty is the fabrication of such devices using a simple commercial inkjet printer to print a carbon nanotube conducting surface followed by a layer of cyanobacterial cells onto a paper support. This work is motivated by the need of sources for low-power applications such as disposable and biodegradable biosensors. According to the authors the power requirements for such applications are between 10 and 100 W with a voltage in the range of 1-2 V.

While the authors succeed in demonstrating a scalable processing approach that involves the printing of the different layers directly on paper, the overall performance level that is achieved (maximum power of 27 microwatt/square meter with a voltage of a few mV) remains orders of magnitude below the minimum requirements for applications. Having a device with an active area of several square meters is just not practical. Furthermore, the overall voltage produced per device is very low.

Since the main novelty of this work resides in the demonstration of devices using processing techniques that are scalable, the severe penalty in performance that is observed when deviating from previous approaches leaves the reader with the impression that the proposed approach is not that successful. Hence, the significance of this work can be considered as low. Therefore, I do not think that this work is news worthy as I would not consider it a significant step forward in this field at this stage.

Reviewer #1 (Remarks to the Author)

1. Summary of the key results

The study presents a new design of paper-based BPV cell, which is able to digitally print live cyanobacterial cells on the electrode surface and generates electrical current. The formation of a solid culture is different from traditional liquid culture.

2. Originality and interest: if not novel, please give references

While the concept is interesting, and novelty of the study can be better justified. The inkjet method for bacteria deposition has been reported before, and it is not clear what's new in the deposition method for cyanobacteria in this study. An example study is "Xu, et al. Construction of high density bacterial colony arrays and patterns by the ink-jet method. Biotechnology and Bioengineering, 2004, DOI: 10.1002/bit.10768).

Response

The novelty of our paper does not lie in simply depositing cyanobacterial cells using inkjet printing, but in doing so in a way that actually allows the printed cells to produce electrical power. To emphasise this, we have included substantial amounts of new data in the revised manuscript to support the feasibility of using printed cyanobacteria to drive low-power devices (see Figure 4). We have also rewritten the manuscript to better emphasise the novel aspects of our work. While it is certainly true that Xu et al. (2004) described the use of an ink-jet printer to print bacterial cells, their work was restricted to printing cells onto small-scale agar-coated glass cover slips. Our work goes further as we demonstrate that we are able to print cyanobacteria onto ordinary paper, a non-standard, relatively cheap culture substrate with greater flexibility for scalability and industrial applications, and that the resulting printed cells are still able to produce an electric current, something that has never been investigated before.

In terms of novelty in the area of converting solar energy to electric current, we feel that the generation of electric current by printed cyanobacteria

complements very nicely other recent approaches considered to be suitable for publication in *Nature Communications* such as the recent work of Pinhassi *et al.* (2016) *Nat. Comm.* 7, 12552, DOI: 10.1038/ncomms12552, which reported the use of isolated thylakoids rather than live cells to generate an electric current. However, our approach extends their work on several fronts: our BPV cell is a compact semi-dry thin-film device that does not require chemical mediators; the bioelectrode is self-repairing and hence a more stable system, and overall the BPV cell is a more realistic proposition for scale-up.

Similarly, using bioelectrochemical systems for biophotovoltaic applications have been reported and recently reviewed, as the author mentioned. (See McCormick et al., Biophotovoltaics: oxygenic photosynthetic organisms in the world of bioelectrochemical systems, Energy & Environmental Science, 2015, DOI: 10.1039/C4EE03875D). It is not clear what advantages this study has and why using solid culture is novel, applicable, and sustainable. More specifics are needed.

Response

In response to the reviewer's helpful comment, we have substantially revised the Introduction in order to clarify the importance of this work and the gaps in knowledge that it is addressing.

In summary, our study describes three important advances in the fields of biophotovoltaic and extracellular electron transport research. 1) We demonstrate that cyanobacterial cells can be printed in a viable state using an inkjet-printing method, 2) we demonstrate that inkjet printing can be used to fabricate both the non-biological and biological parts of a bioelectrode and that this bioelectrode produces an electric current at similar levels to the conventional bioelectrode used in BPV cells, and 3) we describe the fabrication of a novel thin-film semi-dry BPV system (printed bioanode and printed cathode on paper with hydrogel covering). We believe that these three innovations will be important for the miniaturisation and industrial-scale production of BPV cells.

3. Data & methodology: validity of approach, quality of data, quality of presentation

The manuscript mainly discusses the manufacturing process of the biophotovoltaic cells and the process of experiment, but it has very limited information on scientific data reporting, characterization, or discussion. Plus, it is not clear what the research hypothesis is and what scientific questions are answered by the study.

The experimental design and reporting can be improved significantly. For example, it would be very informative if the authors connect different parameters such as light density, biomass growth, and power production and quantify their relationships and discuss how to improve system performance.

Response

In response to the reviewer's comments, we have now performed additional experiments to further characterise our printed system: we show the effect of light intensity on the performance (Figure 3) and present direct evidence of the capability of the bioelectrode to power a digital clock and an LED (Figure 4). Ways to improve the system performance are discussed in the last two paragraphs of the Discussion.

4. Appropriate use of statistics and treatment of uncertainties
ANOVA test was used to characterize the viability of data.

5. Conclusions: robustness, validity, reliability

The conclusions can be better supported by more comprehensive data and characterization. Please see comments above.

Response

As requested by the reviewer, and mentioned earlier in this rebuttal, the revised manuscript has been extensively rewritten to highlight the novelty of the work and now contains substantial amounts of new data to characterize the printed bioelectrode.

6. Suggested improvements: experiments, data for possible revision

Below are some specific comments and suggestions to help improve the quality of the study.

Page 8 - The compact gel to replace liquid reservoir growth media can be interesting, but can you provide data on what rate the gel substrate is consumed, how to replenishing the gel substrate, and why this is better than liquid media?

Response

It is apparent from the reviewer's comment that we failed to explain clearly the role of the hydrogel in the initial submission. It is not a substrate to be consumed but a mechanical support to hold the water in place for the cells. The hydrogel film is equivalent to the container of the liquid media. It has the advantages of compact scale and efficient material use. In the case of conventional BPV cells, the liquid contained in the device is the medium used to grow the cells that were deposited onto the electrode by gravity. Very little of this water is needed for keeping the cells hydrated and viable for photosynthesis. By using a solid culture approach, we remove this excess water from the BPV cell, thereby miniaturising the device. From an industrial point of view, the semi-dry thin-film cell offers several advantages for scale-up, which we now discuss in the revised manuscript in the first paragraph of the Discussion on page 19.

Page 10 - Figure 3. The power outputs are among the lowest reported by miniature microbial fuel cells, presumably due to the extremely large resistance present in the system and limited bacterial growth. Can you elaborate more on how did the configuration and light exposure affect power outputs, what are the limiting factors, and how to improve and sustain system performance?

Response

In the revised manuscript, we now present a comprehensive analysis of the power output of the printed bioelectrode under experimental conditions that allow a direct comparison with conventional bioelectrodes (see Figures 2&3). We

have found that power generation from the printed bioelectrode is comparable to previously published values in the literature obtained with conventional bioelectrodes.

Page 11 - Figure 3. The anode only control shows pretty significant power output relative to the anodes with cells. Can you explain why?

Response

The power output of the control anode is three times smaller than the one with the printed cells (0.07 and 0.22 mWm⁻² respectively). The source of this power output is still unclear but might be related to spurious reactions on the carbon nanotube electrode. Generation of low amounts of power without cells is not uncommon in these devices. However, in contrast to the printed bioelectrode, the power output declines with time and is independent of light.

Page 18 - Can you provide more details on how the bioink was made and maintained?

Response

Details of the preparation of the bioink are explained in Material & Methods on page 22. The bioink was used immediately after pelleting cells by centrifugation and resuspending in medium. The bioink was transferred to a sterile inkjet cartridge by micropipetting and the cartridge was kept in the printer with the printer lid closed. Overall the process took about two hours to transfer cells from liquid culture to a printed solid culture state.

Page 19 - How was the anode and cathode arranged to prevent short circuiting? Any separators applied?

Response

The paper substrate acted as the separator. The design of the device is explained in detail in Figures 2, 5 & 6 of the revised manuscript. We also suggest in the

Discussion the use of insulation such as inkjet-printable hydrophobic polymer to improve the system performance and cite Määttänen *et al.* (2011) Paper-based planar reaction arrays for printed diagnostics. *Sensors Actuators B* **160**, 1404–1412.

Page 21 - the 1M Ω resistor used in the study seems too big, because it significantly limits electron transfer between the anode and cathode. Most MFC studies use 10-1000 Ω to allow bacteria to transfer electrons.

Response

It is due to the high resistance of the commercially available carbon nanotube ink used (Nink-1000: multiwall, NanoLab, USA). We also encountered batch-to-batch variation in the resistivity with this material, which led to a higher resistance in the printed BPV system which consists of both printed anode and cathode, compared to the hybrid system (Figure 2). This point is now clarified in the Results and Discussion.

7. References: appropriate credit to previous work?

The authors provided appropriate credits to previous work.

8. Clarity and context: lucidity of abstract/summary, appropriateness of abstract, introduction and conclusions

The writing is understandable, but really the science is missing.

Response

We have rewritten the revised manuscript to emphasise the novelty of our work.

Reviewer #2 (Remarks to the Author)

This paper deals with biophotovoltaic systems and more precisely with the demonstration of a device that generates electrical power from microbacteria.

The main novelty is the fabrication of such devices using a simple commercial

inkjet printer to print a carbon nanotube conducting surface followed by a layer of cyanobacterial cells onto a paper support. This work is motivated by the need of sources for low-power applications such as disposable and biodegradable biosensors. According to the authors the power requirements for such applications are between 10 and 100 W with a voltage in the range of 1-2 V.

While the authors succeed in demonstrating a scalable processing approach that involves the printing of the different layers directly on paper, the overall performance level that is achieved (maximum power of 27 microwatt/square meter with a voltage of a few mV) remains orders of magnitude below the minimum requirements for applications. Having a device with an active area of several square meters is just not practical. Furthermore, the overall voltage produced per device is very low.

Since the main novelty of this work resides in the demonstration of devices using processing techniques that are scalable, the severe penalty in performance that is observed when deviating from previous approaches leaves the reader with the impression that the proposed approach is not that successful. Hence, the significance of this work can be considered as low. Therefore, I do not think that this work is news worthy as I would not consider it a significant step forward in this field at this stage.

Response

We thank the reviewer for highlighting the need to discuss more rigorously the power output and scalability of our printed bioelectrode. In the revised manuscript, we present new experiments to demonstrate that a printed bioelectrode covering a total area of half an A4 piece of paper (not “several square metres” as suggested by Reviewer 2) is capable of powering a digital clock (Figure 4a, b) or an LED (Figure 4c, d). This means that the printed BPV system is capable of both sustained low-power and generating short bursts of relatively high power; both are requirements for real-life applications, such as, for instance, environmental monitoring biosensors. This is quantified in the text (use between 10 and 100 μ W at 1-2 V) on page 20. We therefore believe that our new results are a significant advance on the conventional BPV system in terms of scalability.

Electricity Generation from Digitally Printed Cyanobacteria

(Note that the 'micro' symbol appears to have been lost either from the reviewer's comment 'power requirements are between 10 and 100 W..' or from the version of the manuscript received by the reviewer. Our original referred to 10 μ W and 100 μ W).

Reviewers' comments:

Reviewer #1 (Remarks to the Author):

The authors did a good job adding new data and discussions on the study and provided thorough responses on the concerns raised by the reviewers. I think they made very good progress in improving the quality of the study and manuscript. I have the following comments for them to consider:

Introduction: Can you elaborate more on why BPV application is promising? A major benefit of using microbial fuel cells to generate electricity is its associated wastewater treatment. For BPV, it doesn't have that function rather it competes with low cost and very fast abiotic PV process on electricity generation from sunlight. Some insights on the application potential of this technology will be important.

Figure 2d: Can you elucidate the electricity generation mechanisms of *Synechocystis* – especially explain why similar current was produced in the dark and in the light – this is not very common based on photosynthetic bacterial electron transfer process, so more discussion is needed. It will also be helpful to compare this with Figure 3a, which clearly shows a light-dark cycle.

Figure 5 – We have used hydrogel before in microbial fuel cell work, but we found the hydrogel show significant volume change during wet-dry cycle, which led to connection issues when dry (shrink and causes disconnection). Did you find similar issues? If so, how did you overcome it?

Response to referee 1

The authors did a good job adding new data and discussions on the study and provided thorough responses on the concerns raised by the reviewers. I think they made very good progress in improving the quality of the study and manuscript. I have the following comments for them to consider:

Introduction: Can you elaborate more on why BPV application is promising? A major benefit of using microbial fuel cells to generate electricity is its associated wastewater treatment. For BPV, it doesn't have that function rather it competes with low cost and very fast abiotic PV process on electricity generation from sunlight. Some insights on the application potential of this technology will be important.

Response: A direct comparison between BPV and conventional PV is at the moment difficult to make because BPV technology is still in its infancy and because BPV and PV are based on different principles. BPV devices are in effect fuel cells and therefore subject to different constraints to PV. One important advantage of BPV over PV systems is that BPVs are capable of producing electricity in the absence of light by consuming products of photosynthesis that accumulated previously in the light. This point is now highlighted in the Introduction on page 3.

The potential use of BPV as an environmentally friendly power supply for low power applications is also emphasised on page 3. We then discuss potential applications in more detail in the Discussion on pages 20-21 (text repeated below).

'The paper-based thin-film BPV cell might also form the basis of a disposable and environmentally friendly power supply for use in paper-based analytical devices (PADs), which have attracted considerable attention for point of care applications by combining the advantages of low cost and ease of use with sensitivity, specificity, robustness and disposability⁴³⁻⁴⁵. We can therefore envision future applications where PADs, disposable electronics and paper-based thin-film BPV power supplies are fully integrated into a single biodegradable paper-based lab-on-a-chip.'

*Figure 2d: Can you elucidate the electricity generation mechanisms of *Synechocystis* – especially explain why similar current was produced in the dark and in the light – this is not very common based on photosynthetic bacterial electron transfer process, so more discussion is needed. It will also be helpful to compare this with Figure 3a, which clearly shows a light-dark cycle.*

Response:

In both Fig. 2 and Fig. 3, the comparison of current production between the light and dark shows the light current being higher. We have clarified this with additional text on page 10.

To further clarify the source of the light and dark currents, original text in the Discussion on pages 19-20 has been modified.

As with all photosynthetic organisms, electricity is generated in *Synechocystis* by a combination of photosynthesis, respiration and fermentation. In the presence of light, electron flow from the photosynthetic reactions will dominate, while in the dark the products of carbon fixation driven by the photosynthetic light reactions will be metabolised (by respiration or fermentation) producing an electron flow. Therefore, for all photosynthetic organisms, currents are recorded both in the dark and in the light without the addition of an external source of carbon. In all cases the magnitude of the current measured in the dark is smaller than the one measured in the light.

Figure 5 – We have used hydrogel before in microbial fuel cell work, but we found the hydrogel show significant volume change during wet-dry cycle, which led to connection issues when dry (shrink and causes disconnection). Did you find similar issues? If so, how did you overcome it?

Response: We have experienced similar effects due to evaporation. For this reason, the experiments with the hydrogel were carried out in an environment with controlled humidity as detailed in the Methods section on page 26 and explained on page 17. We also have clarified on page 17 the distinct roles played by hydrogels in MFCs and in the thin-film BPV device. In the Discussion page 21, we also suggest that a gas-permeable membrane could be easily placed on top of the hydrogel to limit evaporation.

REVIEWERS' COMMENTS:

Reviewer #1 (Remarks to the Author):

I think the authors made sufficient edits to address the comments, and I am satisfied with the current version.